# Efficient Machine Learning Models for Early Stage Detection of Autism Spectrum Disorder

**Mousumi Bala** [1], **Mohammad Hanif Ali** [2], **Md. Shahriare Satu** [3], **Khondokar Fida Hasan** [4] **and Mohammad Ali Moni** [5,*]

1 Department of Computer Science and Engineering, Eastern University, Ashulia Model Town, Savar 1345, Bangladesh; mousumi.cse@easternuni.edu.bd
2 Department of Computer Science and Engineering, Jahangirnagar University, Savar 1342, Bangladesh; hanif_ju03@juniv.edu
3 Department of Management Information Systems, Noakhali Science and Technology University, Sonapur 3814, Bangladesh; shahriarsetu.mis@nstu.edu.bd
4 Centre for Cyber Security Research & Innovation, RMIT University, 124 La Trobe Street, Melbourne, VIC 3000, Australia; fida.hasan@rmit.edu.au
5 Artificial Intelligence & Digital Health, School of Health and Rehabilitation Sciences, Faculty of Health and Behavioural Sciences, The University of Queensland, St. Lucia, QLD 4072, Australia
* Correspondence: m.moni@uq.edu.au

**Abstract:** Autism spectrum disorder (ASD) is a neurodevelopmental disorder that severely impairs an individual's cognitive, linguistic, object recognition, communication, and social abilities. This situation is not treatable, although early detection of ASD can assist to diagnose and take proper steps for mitigating its effect. Using various artificial intelligence (AI) techniques, ASD can be detected an at earlier stage than with traditional methods. The aim of this study was to propose a machine learning model that investigates ASD data of different age levels and to identify ASD more accurately. In this work, we gathered ASD datasets of toddlers, children, adolescents, and adults and used several feature selection techniques. Then, different classifiers were applied into these datasets, and we assessed their performance with evaluation metrics including predictive accuracy, kappa statistics, the f1-measure, and AUROC. In addition, we analyzed the performance of individual classifiers using a non-parametric statistical significant test. For the toddler, child, adolescent, and adult datasets, we found that Support Vector Machine (SVM) performed better than other classifiers where we gained 97.82% accuracy for the RIPPER-based toddler subset; 99.61% accuracy for the Correlation-based feature selection (CFS) and Boruta CFS intersect (BIC) method-based child subset; 95.87% accuracy for the Boruta-based adolescent subset; and 96.82% accuracy for the CFS-based adult subset. Then, we applied the Shapley Additive Explanations (SHAP) method into different feature subsets, which gained the highest accuracy and ranked their features based on the analysis.

**Keywords:** ASD; machine learning; classifier; feature selection; prediction model

## 1. Introduction

Autism spectrum disorder (ASD) is a neuro-developmental disorder where it appears in human beings during the first three years [1]. It is basically characterized by several symptoms such as impairments in social interaction, communication, restricted interests, and repetitive behavior [2]. Individuals with ASD face difficulty understanding other's feelings and thinking. They experience many problems communicating with others. As reported by the World Health Organization (WHO), throughout the world, around 1 in 270 individuals has ASD [3]. Each individual with ASD has unique characteristics, and some have exceptional abilities in visual, academic, and music skills. In this case, the most important steps are required to detect ASD and to ensure proper treatment as early as possible. These steps are helpful to decrease the effects of this disorder and to improve

their condition. The symptoms of ASD are identified by different types of observations. However, a significant amount of time and effort are needed where early detection is useful for providing better treatment for ASD patients. Recently, machine learning methods are widely used for analyzing the symptoms of various severe diseases like heart disease, diabetes, and cancer tissues, etc. Therefore, many researchers explored numerous methods [4,5] that are helpful to identify ASD patients and decrease the affects of ASD more precisely.

The objective of this work was to propose a machine learning model that explores ASD of toddlers, children, adolescents, and adults at an early stage as well as to investigate the individual characteristics of them more efficiently. In this model, we generated several feature subsets of toddlers, children, adolescents, and adults using various feature selection techniques. Different classification methods were applied into primary and its feature subsets. Then, their results were compared, and we determined the best classifier for each age group and the feature subset for which the highest results were obtained. Additionally, the performance of these classifiers were investigated using a non-parametric statistical significant test. Then, we interpreted the results of the best feature subsets and selected significant features of ASD using an explainable AI method. The following concise summary of the contribution is given as follows:

- We proposed an efficient machine learning method that has the potential to identify ASD with high accuracy at an early stage.
- We concentrated on identifying important feature subsets and explained different features to know how individual features are responsible to generate the best result to diagnose ASD or not.
- To justify the performance of the classifier, we used a non-parametric statistical method and checked the classifier's pairwise significance.
- This method is helpful to identify ASD in a simple and flexible way.

This article is structured as follows: Section 2 contains a literature review on ASD screening approaches. Section 3 represents the working steps of detecting ASD and its characteristics. Section 4 describes the experimental results and an interpretation of these results. Section 5 describes the discussion and conclusion.

## 2. Literature Review

Many state-of-art works were happened to investigate, classify and explore significant factors of ASD. Thabtah et al. [6–9] developed a mobile application named ASDTests for data collection related to ASD for toddlers, children, adolescents, and adults. This app was built based on Q-CHAT and AQ-10 tools to predict ASD or not. They collected ASD data using this app and uploaded them into the University of California-Irvine (UCI) Machine Learning (ML) repository. Omar et al. [10] proposed an effective machine learning model where they analyzed AQ-10 and 250 real datasets with Random Forest (RF), Classification and Regression Trees (CART) and Random Forest-Iterative Dichotomiser 3 (ID3). Sharma et al. [11] investigated these datasets by employing CFS-greedy stepwise feature selector and further applied Naïve Bayes (NB), Stochastic Gradient Descent (SGD), K-Nearest Neighbours (KNN), Random Tree (RT), and K-Star (KS) into these datasets. Satu et al. [12] collected some samples of 16–30 years children where several tree based classifiers were used to investigate them and extracted several rules for normal and autism. Erkan et al. [13] analyzed similar datasets by implementing KNN, SVM, and RF where RF showed the best performance to identify ASD. Another study by Thabtah et al. [14] generated several feature subsets of adults and adolescents using Information Gain (IG) and Chi-Squared (CHI) where Logistic Regression (LR) was used to identify ASD from them. Akter et al. [15] gathered toddlers, children, adolescents, and adults datasets and generated some transformed sets. Then, different classifiers were used to analyze them where SVM showed the best performance for the toddler as well as Adaboost provided both children and adult. In addition, Glmboost showed the best outcomes the adolescent dataset. Hossain et al. [16] investigated similar types of datasets and generated subsets using CFS,

CHI, IG, One-R and Relief-F methods. Further, they employed LR, Multilayer Perceptron (MLP), Sequential minimal optimization (SMO) into them. Then, SMO showed the best accuracy 91% for child, 99.9% for adolescent, 97.58% for adult datasets. Raj et al. [17] scrutinized these datasets (i.e., excluding toddler) using SVM, LR, NB, and Convolutional Neural Network (CNN) where CNN showed the highest accuracy 98.30% for child, 96.88% for adolescent, and 99.53%, for adult, respectively. Again, Thabtah et al. [18] developed a Rules-based Machine Learning (RML) method for extracting ASD traits where RML gave higher predictive accuracy from other machine learning approaches. Chowdhury et al. [19] provided an association classification technique with seven algorithms where this method showed 97% accuracy to detect ASD. Akter et al. [20] gathered ASD dataset of different age levels and generated some transformed subsets of it. They analyzed them with several classifiers where LR outperformed other classifiers and extracted significant traits. Akter et al. [21] also extracted several autism subtypes using k-means algorithms. Then, they identified different discriminatory factors among them.

## 3. Methodology

In this work, we used different feature selection methods and generated some feature subsets. Then, some classification algorithms were applied into primary toddler, child, adolescent, and adult datasets and their feature subsets. The performance of different classifiers were investigated to determine which features are more useful to detect ASD from controls. Figure 1 shows the proposed model of ASD detection at early stage.

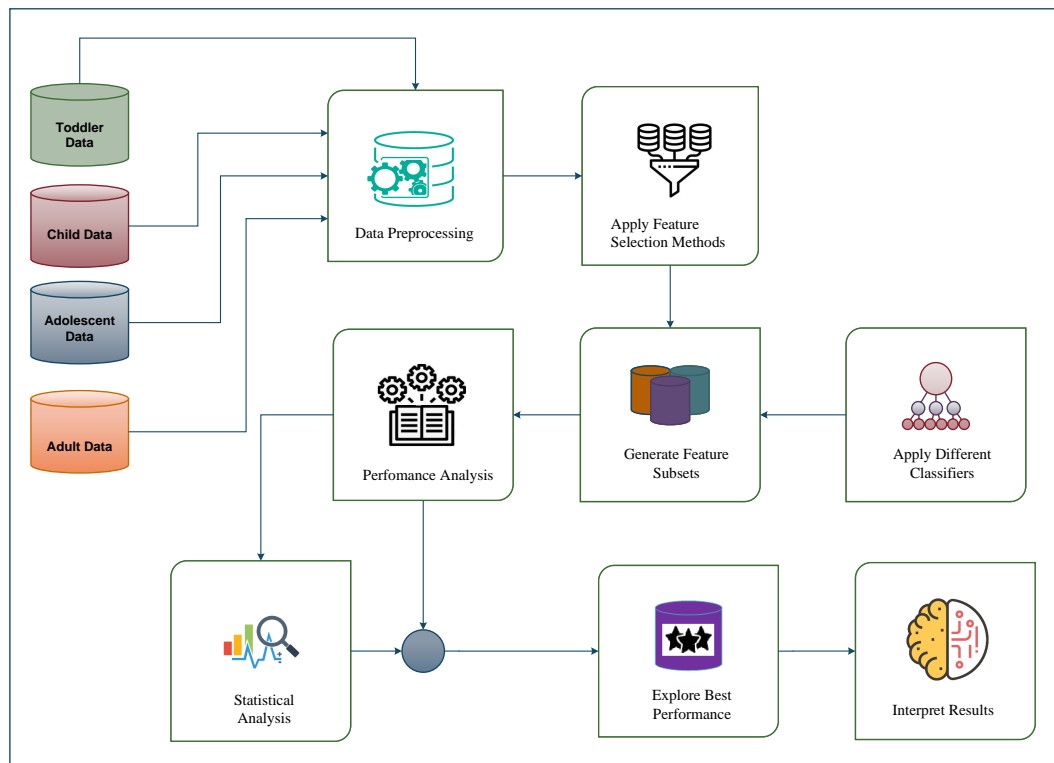

**Figure 1.** The proposed framework for early ASD detection.

### 3.1. Data Preprocessing

**Dataset Description:** Thabtah et al. [6] from the Nelson Marlborough Institute of Technology developed an autism screening app called ASDTests, which was used for data collection from the target audience (toddlers, children, youths, and adults). It used Q-CHAT-10 and AQ-10 questionnaires (AQ-10 Child, AQ-10 Adolescent, and AQ-10 Adult) to determine ASD risk factors. This app automatically computed its final-score from 0 to 10. It indicates a positive prediction of ASD if the final-score is greater than 6 out of 10. In this work, we used toddler, child, adolescent, and adult ASD datasets (version 2) [22–25],

where the toddler dataset consisted of 18 features; however, the child, adolescent, and adult datasets contained 23 features. These datasets contained the records of 12–36 months (toddlers), 4–11 years (child), 12–16 years (adolescent), and 18 years or greater (adult) age groups. Table 1 shows the details of these datasets, and Table 2 provides the description of individual features of these datasets that were used for analysis.

**Table 1.** Data description.

| Dataset Name | Instances | Attributes | Male/Female | Age (Years) | Average Age | ASD/Normal |
|---|---|---|---|---|---|---|
| Toddler Autism dataset | 1054 | 18 | 735/319 | 12–36 (mons) | 27.86 (mons) | 735/319 |
| Autism Child Dataset (Version-2) | 509 | 23 | 363/146 | 11-Apr | 6.39 | 257/252 |
| Autism Adolescent Dataset (Version-2) | 248 | 23 | 117/131 | 16-Dec | 14.04 | 127/121 |
| Autism Adult Dataset (Version-2) | 1118 | 23 | 596/522 | 17–80 | 30.143 | 358/760 |

**Table 2.** Features description.

| Feature | Type | Description |
|---|---|---|
| Age (Year) | Number | Toddlers (months), children, adolescent, and adults (year) |
| Gender | String | Male or female |
| Ethnicity | String | List of common ethnicities |
| Born with jaundice | Boolean | Whether the case was born with jaundice |
| Family member with PDD | Boolean | Whether any immediate family member has a PDD history |
| Who is completing the test | String | Parent, self, caregiver, medical staff, clinician, etc. |
| Country of residence | String | List of countries |
| Used the screening app before | Boolean | Whether the user has formerly used the screening app |
| Language | String | The user's language details |
| A1: Answer of Q1 | Binary(0,1) | Does your child look at you when you call his/her name? (Toddler) S/he often notices small sounds when others do not. (Child) S/he notices patterns in things all the time. (Adolescent) I often notice small sounds when others do not. (Adult) |
| A2: Answer of Q2 | Binary(0,1) | How easy is it for you to get eye contact with your child? (Toddler) S/he usually concentrates more on the whole picture rather than the small details. (Child, adolescent, adult) |
| A3: Answer of Q3 | Binary(0,1) | Does your child point to indicate that s/he wants something? (Toddler) In a social group, s/he can easily keep track of several different people's conversations. (Child) In a social group, s/he can easily keep track of several different people's conversations. (Adolescent) I find it easy to do more than one thing at once. (Adult) |
| A4: Answer of Q4 | Binary(0,1) | Does your child point to share interest with you? (Toddler) S/he finds it easy to go back and forth between different activities. (Child) If there is an interruption, s/he can switch back to what s/he was doing very quickly. (Adolescent) If there is an interruption, I can switch back to what I was doing very quickly (Adult) |
| A5: Answer of Q5 | Binary(0,1) | Does your child pretend? (Toddler) S/he does not know how to keep a conversation going with his/her peers. (Child, Adolescent) I find it easy to "read between the lines" when someone is talking to me. (Adult) |
| A6: Answer of Q6 | Binary(0,1) | Does your child follow where you are looking? (Toddler) S/he is good at social chit-chat (Child, Adolescent). I know how to tell if someone listening to me is getting bored (Adult) |
| A7: Answer of Q7 | Binary(0,1) | If you or someone else in the family is visibly upset, does your child show signs of wanting to comfort them? (Toddler) When s/he is read a story, s/he finds it difficult to work out the character's intentions or feelings (Child). When s/he was younger, s/he used to enjoy playing games involving pretending with other children (Adolescent). When I am reading a story I find it difficult to work out the characters' intentions (Adult). |

**Table 2.** *Cont.*

| Feature | Type | Description |
|---------|------|-------------|
| A8: Answer of Q8 | Binary(0,1) | Would you describe your child's first words as: (Toddler) When s/he was in preschool, s/he used to enjoy playing pretending games with other children (Child). S/he finds it difficult to imagine what it would be like to be someone else (Adolescent). I like to collect information about categories of things (Adult). |
| A9: Answer of Q9 | Binary(0,1) | Does your child use simple gestures? (Toddler). S/he finds it easy to work out what someone is thinking or feeling just by looking at their face (Child). S/he finds social situations easy (Adolescent). I find it easy to work out what someone is thinking or feeling just by looking at their face (Adult). |
| A10: Answer of Q10 | Binary(0,1) | Does your child stare at nothing with no apparent purpose? (Toddler). S/he finds it hard to make new friends (Child, Adolescent). I find it difficult to work out people's intentions (Adult). |
| Screening score | Integer | It is based on the scoring algorithm of the screening method used |
| Class | String | ASD or No ASD |

**Data Cleaning:** We cleaned these datasets to simplify this model and increased classification accuracy by deleting instances with missing values. After that, we discarded some irrelevant features (i.e., those which are not related to ASD) such as the case, whether they the used app before, the user (who completed the screening), the language, why they had taken the screening, the age description, the screening type, and the score, respectively. The score 7 to 10 was used to classify ASD prediction for the child, adolescent, and adult datasets. A value of 4 or higher was classified as ASD for the toddler dataset. In this work, we selected 16 features for the child, adolescent, and adult datasets and 15 features for the toddler dataset, respectively.

*3.2. Implementing Feature Selection Methods*

Feature selection is required to identify significant attributes that improve the performance of machine learning models [26]. Many state-of-the-art works were found where various significant features were identified to detect autism more efficiently [27] On the toddler, child, adolescent, and adult datasets, various feature selection techniques such as the Boruta algorithm, Correlation-based Feature Selection with Harmony Search (CFS–Harmony Search), Repeated Incremental Pruning to Produce Error Reduction (RIPPER), and Recursive Feature Elimination (RFE) were used to explore various feature subsets. A brief description about them is given as follows:

- **Boruta algorithm** is a wrapper algorithm based on the random forest [28] where it finds the importance of a feature by creating shadow features [29]. It is an extended system where each feature of the given data set is replicated. Then, the values of replicated variables are randomly combined, which are called shadow features. It performs its feature selection process using RF on the extended data set and evaluates the importance of each feature. Additionally, it computes the z-score of real and shadow features. It compares higher z-score values of real features than the maximum z-score value of its shadow features at every iteration. In this process, it constantly eliminates features that are deemed highly unimportant. Finally, the algorithm ends either when all features are approved or rejected or it obtains a particular limit of RF runs [30].

- **Correlation-based Feature Selection (CFS–Harmony Search)** is a heuristic function that evaluates the ranks of features based on their correlation [31]. For harmony search, it illustrates the point of intersection with the following parameters: the number of harmonies in memory $N$, the number of indicator $M$, the number of possible values of indicator $D$, the number of optimal indicator $i$ in the harmony memory $= E_i$, and the rate of harmony memory $E_t$. The probability is calculated by the following equation,

$$P_t(E) \subseteq_{i=1}^{M} [Et\frac{Ei}{N} + (1 - Et)\frac{1}{D}] \tag{1}$$

- **Repeated Incremental Pruning to Produce Error Reduction (RIPPER)** is the rule induction algorithm that was introduced by W. Cohen in 1995 [32]. It is generated as a set of if-then-else rules that evolves several iterations of the rule learning algorithm. This model is maintained by three-steps such as grow, prune, and optimize [33] where the evaluation process is denoted as,

$$Gain(R_0, R_1) = c[log(\frac{T1}{T1 + F1}) - log(\frac{T0}{T0 + F0})] \tag{2}$$

  where $R_0$ = the initial rule, $R_1$ = the rule after adding conjunct, $c$ = the number of true instances covered by $R_0$ and $R_1$, $T_0$ = the number of true instances covered by $R_0$, $F_0$ = the number of false instances covered by $R_0$, $T_1$ = the number of true instances covered by $R_1$, and $F_1$ = the number of false instances covered by $R_1$ [34].

- **Recursive Feature Elimination (RFE)** is a feature selection technique that discards the least important features recursively. In this process, the initial features are trained where each important feature is acquired through any selected attributes [35]. Then, the least important features are eliminated from the initial feature set. This process is recursively repeated until the desired number of feature subsets are obtained [36]. The steps of RFE are given as follows [37]:

  1. Train the classifier.
  2. Calculate the score for all features with the ranking.
  3. Eliminate the feature with the lowest score.

### 3.3. Apply Individual Classification Methods

After generating various feature subsets, 30 widely used classifiers were implemented into all of these datasets and their subsets. Some classifiers that produced less than 70% accuracy were eliminated. Thus, the performance of NB, KS, C4.5, CART, SVM, KNN, Bagging (BG), and Random Tree (RT) were considered. Then, we compared their performance and detected the best classifier. In addition, it detects feature subsets for which the classifier shows the highest outcomes. Therefore, a brief description of them are given as follows:

- **Naïve Bayes (NB)** is a most constructive probabilistic classifier based on Bayes theorem [38]. It predicts target output more efficiently according to the foundation of the probability of an entity. The formula of Bayes theorem is given as,

$$R(a|b) = \frac{R(b|a)R(a)}{R(b)} \tag{3}$$

  The derivation form for NB is:

$$R(a|b) = R(b_1|a) * R(b_2|a) * \dots * R(b_n|a) * R(a) \tag{4}$$

  Here, $R(a \mid b)$ is the subsequent probability of the target class; $R(a)$ is the earlier probability of the class; $R(b \mid a)$ is the prospect of the predictor specified class; and $R(b)$ is the earlier probability of the predictor (see Equation (3)).

- **K-Star (KS)** is an instance-based classifier that categorizes samples or instances by differentiating it based on pre-categorized samples [39]. Some similar functions are used to determine the class of test instances. It uses an entropy-based function, which differentiates from other instance-based learners [40].

- **Decision Tree (C4.5)** is the extension version of the ID3 algorithm that uses the recursive divide and conquer method to produce the C4.5 decision tree [41]. When unknown data are found, this method predicts a target class by satisfying several conditions. C4.5 uses Information Gain (IG) that calculates the gain ratio by the following equation [42],

$$Gainratio(Attribute) = \frac{Gain(Attribute)}{SpiltInfo(Attribute)} \tag{5}$$

- **Classification and Regression Trees (CART)** use several classification and regression trees where classification trees are used to accumulate the finite number of unsorted values and determine the prediction errors. Besides, regression trees are employed for grouping sorted or ordered values and determining the prediction error by calculating the root squared difference between target and predicted values [43].
- **K-Nearest Neighbour (KNN)** is used for classifying instances based on their nearest neighbors [44]. It generally takes more than one neighbor and determines their distances using the Euclidean method, which is calculated with the following equation [45],

$$D = \sqrt{(m_1 - m_2)^2 + (n_1 - n_2)^2} \tag{6}$$

where, $D$ is distance between $(m_1, n_1)$ and $(m_2, n_2)$ points.
- **Support Vector Machine (SVM)** generates some vectors to create a decision boundary that separates n-dimensional space into classes. This decision boundary is called a hyperplane. In the general situation, two parallel hyperplanes are generated, which concurrently minimizes the classification error and maximizes the margin of classes. It is called a maximum margin classifier [46].
- **Bagging Classifier (BG)** is a parallel ensemble method that generates several random subsets from substitution of the original dataset. Then, we analyzed them using the base classifier and aggregated their predictions by voting [47]. It decreases the variance and correctly predicts the target outcome.
- **Random Tree (RT)** is a decision tree where a set of possible trees are randomly generated with $K$ random features. The combination of large sets of random trees is generally produced with accurate predictions more efficiently [48].

### 3.4. Use Evaluation Metrics for Performance Analysis of Classifiers

The performance measurement is essential for evaluating how well a classification model correctly predicts instances and achieves a desired target [17,49]. The confusion matrix provides a more detailed overview of a predictive model's performance. It represents which classes are being predicted correctly and incorrectly and shows the measurement of type errors. In the confusion matrix, every instance in a given dataset falls into one of the four categories: True Positive (TP), True Negative (TN), False Positive (FP), and False Negative (FN) [16]. The above four category data are arranged in a matrix known as the confusion matrix. Table 3 shows the elements of the confusion matrix.

**Table 3.** Confusion matrix for ASD.

| Target | Predicted | |
| --- | --- | --- |
| | **ASD** | **No ASD** |
| ASD | TP | FN |
| No ASD | FP | TN |

In this work, we used several evaluation metrics including accuracy, kappa statistics, the F1-Score, and AUROC to assess the performance of each classifier. To calculate these metrics, the confusion matrix is required to generate and gather different types of instances from it. These metrics are described briefly as follows:

- **Accuracy:** It is a measure of how effective the model is used to predict outcomes [49], in terms of the total number of predictions:

$$Accuracy = \frac{(TP + TN)}{(TP + FP + FN + TN)} \tag{7}$$

- **Kappa statistics (Kp):** It measures observer agreement for categorical data and expected accuracy and has received considerable attention [50].

$$Kp = \frac{1 - (1 - p_0)}{(1 - p_e)} \tag{8}$$

- **Precision:** It is a measure of true-positive predictions against all retrieved positive instances [51].

$$Precision = \frac{TP}{(TP + FP)} \tag{9}$$

- **Recall:** It is a measure of correctly predicted positive observations against all relevant positive classes [51].

$$Recall = \frac{TP}{(TP + FN)} \tag{10}$$

- **F1 score:** It is the harmonic average of precision and recall [52]:

$$F1 = 2\frac{(Recall * Precision)}{(Recall + Precision)} \tag{11}$$

- **AUROC:** It determines how well true-positive values are isolated from false-positive values [15]:

$$TRP = \frac{TP}{(TP + FN)} \tag{12}$$

$$FRP = \frac{FP}{(FP + TN)} \tag{13}$$

### 3.5. Determining the Performance of Classifiers Using Statistical Tests

After the classification process, we needed to justify these outcomes using various statistical methods and recheck their performance. In this work, the Wilcoxon Signed-Rank (WSR) method was used to test the statistical significance of the individual classifier. We employed this method into the outcomes of different evaluation metrics in the individual age group. A brief description of the WSR method is given as follows:

- **Wilcoxon Signed-Rank Test** is a non-parametric statistical test that is used to compare two independent samples. This method is considered an alternative of the *t*-test when the population mean is not of interest. The working formula of this method is given as follows:

$$W = \sum_{i=1}^{N} [sgn(x_{2i} - x_{1i}).R_i] \tag{14}$$

where $W$ denotes test statistics; $N$ indicates the sample size; $sgn$ denotes a sign function; both $x_{1i}, x_{2i}$ represent the ranked pairs of the two distributions; and $R_i$ indicates the rank.

### 3.6. Interpretation of the Results of Machine Learning Models

Explainable AI is a combination of methods that allows individual users to comprehend the results of machine learning models. In this work, we explored the best performing classifiers which generate the highest results for different feature subsets in the individual age groups. To interpret these outcomes, the SHapley Additive exPlanations (SHAP) method was used to explain which feature vectors are required to generate these predictions. The SHAP method is described in brief as follows:

- **The SHAP method** is a game-theoretic process to explain the output of any individual model. This model was developed by Lundberg and Lee [53]. The purpose of this method is to compute the contribution of each feature for an instance's prediction. It associates optimal credit allocation with a local explanation using Shapley values. The simplest general SHAP values are represented as follows [54]:

  1. Select an objective feature function.
  2. Calculate the Shapley value for all features.
  3. Choose the highest-ranking features.

## 4. Experimental Results

In the primary stage, this study used the Classification and Regression Training (CARET) package in R for feature selection and classification tasks [55]. However, Boruta, CFS, RIPPER, and RFE methods were implemented to produce numerous feature subsets for different age groups (i.e., toddler, child, adolescent, and adult). Table 4 shows the details of these feature subsets, respectively. Then, different baselines (i.e., primary data) along with their feature subsets were scrutinized by eight classifiers such as NB, BG, CART, KNN, C4.5, KS, SVM, and RT, respectively. In this case, we considered the k-fold cross-validation technique for classification, where the value of k is regarded as 10. After the classification process, a non-parametric statistical WSR test was employed to evaluate the performance of individual classifiers using Knowledge Extraction based on Evolutionary Learning (KEEL) software [56]. Then, SHAP summary plots were generated using the shap package in Python for the best classifier and different subsets. Figure 2 shows the details of illustrations about how to interpret machine learning models and explore importance features for generating significant outputs.

**Table 4.** The generated feature subsets of individual age groups.

|  |  | FS Method | Selected Features |
|---|---|---|---|
| Toddler | $FS_{bor}$ | Boruta | A1, A2, A3, A4, A5, A6, A7, A8, A9, A10, Age, Jaundice, Family_ASD |
|  | $FS_{cfs}$ | CFS | A1, A2, A4, A5, A6, A7, A8, A9, A10, Age, User |
|  | $FS_{rpr}$ | RIPPER | A1, A2, A3, A4, A5, A6, A7, A8, A9, A10, Age, Jaundice |
|  | $FS_{rfe}$ | RFE | A1, A2, A3, A4, A6, A7, A8, A9, A10, Gender, User |
|  | $FS_{bic}$ | $FS_{bor}$ Intersect $FS_{rpr}$ | A1, A2, A3, A4, A5, A6, A7, A8, A9, A10, Age |
| Child | $FS_{bor}$ | Boruta | A1, A2, A3, A4, A5, A6, A7, A8, A9, A10, Age, Gender, Residence |
|  | $FS_{cfs}$ | CFS | A1, A2, A3, A4, A5, A6, A7, A8, A9, A10 |
|  | $FS_{rpr}$ | RIPPER | A1, A2, A3, A4, A5, A6, A7, A8, A9, A10, Jaundice |
|  | $FS_{rfe}$ | RFE | A1, A4, A6, A7, A8, A9, A10, Ethnicity, Jaundice, Family_ASD |
|  | $FS_{bic}$ | $FS_{bor}$ Intersect $FS_{cfs}$ | A1, A2, A3, A4, A5, A6, A7, A8, A9, A10 |
| Adolescent | $FS_{bor}$ | Boruta | A1, A2, A3, A4, A5, A6, A7, A8, A9, A10, Age, Ethnicity, Residence, User |
|  | $FS_{cfs}$ | CFS | A1, A2, A3, A4, A5, A6, A7, A10, Ethnicity |
|  | $FS_{rpr}$ | RIPPER | A1, A2, A3, A4, A5, A6, A7, A8, A10, Age, Gender |
|  | $FS_{rfe}$ | RFE | A1, A2, A3, A4, A5, A6, A8, A10, Age, Gender, Ethnicity, Jaundice, Residence |
|  | $FS_{bic}$ | $FS_{bor}$ Intersect $FS_{cfs}$ | A1, A2, A3, A4, A5, A6, A7, A8, A10, Age |
| Adult | $FS_{bor}$ | Boruta | A1, A2, A3, A4, A5, A6, A7, A8, A9, A10, Age, Ethnicity, Residence, User |
|  | $FS_{cfs}$ | CFS | A1, A2, A3, A4, A5, A6, A7, A8, A9, A10, Residence |
|  | $FS_{rpr}$ | RIPPER | A1, A2, A3, A4, A5, A6, A7, A8, A9, A10, Age, Residence, User |
|  | $FS_{rfe}$ | RFE | A1, A2, A4, A6, A7, A8, A9, A10, Age, Residence, Used_App_Before |
|  | $FS_{bic}$ | $FS_{bor}$ Intersect $FS_{rpr}$ | A1, A2, A3, A4, A5, A6, A7, A8, A9, A10, Age, Residence, User |

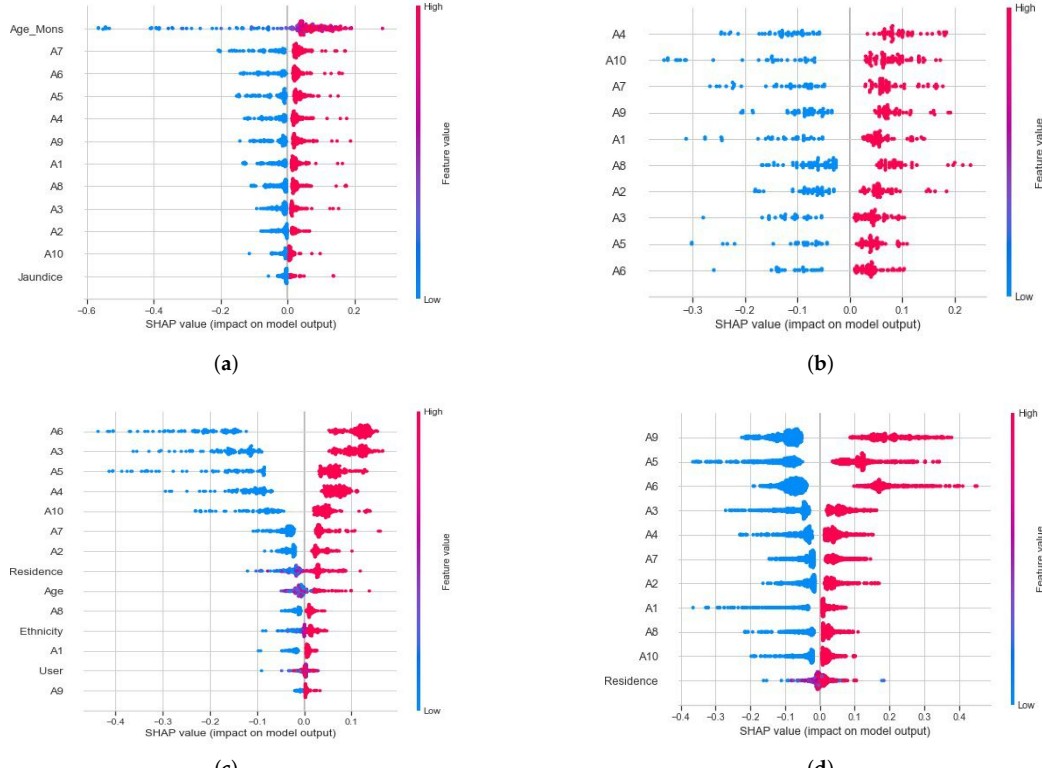

**Figure 2.** Analysis of Shapley values for (**a**) the toddler subset of the RIPPER method, (**b**) the adolescent subset of the Boruta algorithm, (**c**) the child subset of the CFS–Harmony Search method, and (**d**) the adult subset of the CFS method employing the best-performing SVM.

### 4.1. Generating Several Feature Subsets

Different feature selection methods such as Boruta, CFS, RIPPER, and RFE were applied into four ASD baselines and generated $FS_{bor}$, $FS_{cfs}$, $FS_{rpr}$, $FS_{rfe}$, and $FS_{bic}$ for $FS_{bor}$ intersecting with $FS_{cfs}$, respectively [27]. Table 4 shows several feature subsets in different age groups.

### 4.2. Result Analysis of Accuracy

We applied NB, BG, CART, KNN, C4.5, KS, SVM, and RT into four datasets where the highest 96.67%, 95.48%, 95.48%, and 96.06% accuracies were generated by SVM for toddler, child, adolescent, and adult datasets, respectively. After investigating the performance of these classifiers into the baseline and six feature subsets, we found greater improved accuracy to detect autism. In toddlers, SVM gave the maximum accuracy of 97.82% for $FS_{rpr}$ (see Table 5). Then, we observed that the highest accuracy (99.61%) was calculated by SVM for $FS_{cfs}$ and $FS_{bic}$ for children (see Table 6). Regarding the case of adolescents, SVM also provided the maximum accuracy of 95.87% for $FS_{bor}$ (see Table 7). In adults, the highest accuracy (96.82%) was found for $FS_{cfs}$, which was computed by SVM (see Table 8). Besides, other classifiers such as NB, KS, and KNN provided good accuracy similar to SVM. Figure 3a shows the average accuracy of individual classifiers in each age group. Here, we also observed that SVM represented the best average outcomes among all classifiers.

**Table 5.** Performance analysis of classifiers for baseline and its generated feature subsets of toddlers.

| | | NB | BG | CART | KNN | C4.5 | KS | SVM | RT |
|---|---|---|---|---|---|---|---|---|---|
| Accuracy | *Baseline* | 95.44 | 93.16 | 90.89 | 92.69 | 92.31 | 94.21 | 96.67 | 92.12 |
| | $FS_{bor}$ | 95.45 | 94.40 | 92.03 | 94.30 | 92.59 | 95.25 | 97.62 | 92.78 |
| | $FS_{cfs}$ | 96.39 | 92.59 | 91.17 | 94.97 | 91.74 | 95.16 | 96.67 | 92.97 |
| | $FS_{rpr}$ | 95.45 | 94.49 | 92.12 | 95.25 | 92.59 | 95.63 | **97.82** | 92.88 |
| | $FS_{rfe}$ | 95.25 | 93.92 | 92.12 | 94.30 | 92.31 | 95.25 | 95.63 | 93.73 |
| | $FS_{bic}$ | 95.16 | 94.21 | 92.12 | 95.92 | 92.78 | 95.92 | 97.62 | 93.45 |
| Kappa Stat. | *Baseline* | 89.50 | 83.88 | 78.32 | 83.20 | 81.84 | 86.62 | 92.21 | 81.46 |
| | $FS_{bor}$ | 89.55 | 86.71 | 81.09 | 86.90 | 82.47 | 89.08 | 94.43 | 83.18 |
| | $FS_{cfs}$ | 91.59 | 82.41 | 79.15 | 88.38 | 80.60 | 88.78 | 92.20 | 83.46 |
| | $FS_{rpr}$ | 89.85 | 86.92 | 81.30 | 89.08 | 82.47 | 89.96 | **94.87** | 83.44 |
| | $FS_{rfe}$ | 88.80 | 85.62 | 81.55 | 86.96 | 81.84 | 89.01 | 89.54 | 85.52 |
| | $FS_{bic}$ | 88.91 | 86.28 | 81.27 | 90.57 | 82.92 | 90.59 | 94.43 | 84.77 |
| F1-Score | *Baseline* | 95.50 | 93.10 | 90.80 | 92.80 | 92.30 | 94.20 | 96.70 | 92.10 |
| | $FS_{bor}$ | 95.50 | 94.40 | 92.00 | 94.40 | 92.60 | 95.30 | 97.60 | 92.80 |
| | $FS_{cfs}$ | 96.40 | 92.50 | 91.10 | 95.00 | 91.70 | 95.20 | 96.70 | 93.00 |
| | $FS_{rpr}$ | 95.50 | 94.50 | 92.10 | 95.30 | 92.60 | 95.70 | **97.80** | 92.90 |
| | $FS_{rfe}$ | 95.20 | 93.90 | 92.10 | 94.40 | 92.30 | 95.30 | 95.60 | 93.80 |
| | $FS_{bic}$ | 95.20 | 94.20 | 92.10 | 95.90 | 92.70 | 95.90 | 97.60 | 93.50 |
| AUROC | *Baseline* | 99.50 | 98.00 | 91.70 | 94.20 | 92.50 | 98.80 | 96.00 | 90.70 |
| | $FS_{bor}$ | **99.70** | 98.20 | 92.30 | 96.00 | 93.00 | 99.30 | 97.00 | 91.70 |
| | $FS_{cfs}$ | 99.50 | 97.80 | 91.60 | 96.90 | 92.40 | 99.20 | 95.90 | 91.40 |
| | $FS_{rpr}$ | 99.60 | 98.20 | 92.30 | 96.60 | 93.00 | 99.40 | 97.20 | 9.20 |
| | $FS_{rfe}$ | 99.10 | 98.20 | 91.30 | 98.50 | 93.20 | 99.20 | 93.70 | 94.20 |
| | $FS_{bic}$ | 99.60 | 98.20 | 92.40 | 97.30 | 93.30 | 99.50 | 97.00 | 92.60 |

**Table 6.** Performance analysis of classifiers for baseline and its generated feature subsets of children.

| | | NB | BG | CART | KNN | C4.5 | KS | SVM | RT |
|---|---|---|---|---|---|---|---|---|---|
| Accuracy | *Baseline* | 93.12 | 80.94 | 83.71 | 88.61 | 89.39 | 87.22 | 95.48 | 80.74 |
| | $FS_{bor}$ | 94.69 | 81.92 | 83.69 | 92.92 | 89.78 | 93.12 | 95.87 | 78.58 |
| | $FS_{cfs}$ | 93.51 | 90.96 | 90.37 | 94.69 | 90.76 | 95.48 | **99.61** | 92.92 |
| | $FS_{rpr}$ | 93.51 | 90.96 | 89.98 | 92.73 | 89.98 | 93.12 | 98.23 | 90.77 |
| | $FS_{rfe}$ | 88.99 | 86.05 | 87.22 | 89.19 | 88.99 | 89.19 | 91.35 | 85.65 |
| | $FS_{bic}$ | 93.51 | 90.96 | 90.37 | 94.69 | 90.77 | 95.48 | **99.61** | 92.93 |
| Kappa Stat. | *Baseline* | 86.24 | 61.88 | 65.45 | 77.19 | 78.78 | 74.43 | 90.96 | 61.47 |
| | $FS_{bor}$ | 89.38 | 63.85 | 67.39 | 85.85 | 79.57 | 86.23 | 91.74 | 57.13 |
| | $FS_{cfs}$ | 87.02 | 81.92 | 80.75 | 89.38 | 81.53 | 90.95 | **99.21** | 85.85 |
| | $FS_{rpr}$ | 87.02 | 81.92 | 79.96 | 85.45 | 79.96 | 86.23 | 96.46 | 81.53 |
| | $FS_{rfe}$ | 77.99 | 72.10 | 74.46 | 78.38 | 77.99 | 78.37 | 82.71 | 71.31 |
| | $FS_{bic}$ | 87.02 | 81.92 | 80.75 | 89.38 | 81.53 | 90.95 | **99.21** | 85.85 |
| F1-Score | *Baseline* | 93.01 | 80.90 | 82.70 | 88.60 | 89.04 | 87.20 | 95.50 | 80.70 |
| | $FS_{bor}$ | 94.70 | 81.90 | 83.70 | 92.60 | 89.80 | 93.10 | 95.90 | 78.50 |
| | $FS_{cfs}$ | 93.20 | 90.90 | 90.40 | 94.50 | 90.70 | 95.20 | **99.60** | 92.90 |
| | $FS_{rpr}$ | 93.50 | 91.00 | 90.00 | 92.70 | 90.00 | 93.10 | 98.20 | 90.80 |
| | $FS_{rfe}$ | 88.80 | 86.10 | 87.20 | 89.00 | 88.80 | 88.50 | 91.20 | 85.40 |
| | $FS_{bic}$ | 93.50 | 91.00 | 90.04 | 94.70 | 90.80 | 95.50 | **99.60** | 92.90 |
| AUROC | *Baseline* | 98.40 | 88.60 | 85.20 | 90.70 | 91.70 | 95.80 | 95.40 | 81.90 |
| | $FS_{bor}$ | 99.00 | 89.20 | 85.90 | 94.50 | 93.00 | 98.10 | 95.80 | 79.40 |
| | $FS_{cfs}$ | 99.20 | 97.80 | 94.00 | 99.00 | 93.60 | **99.60** | **99.60** | 92.90 |
| | $FS_{rpr}$ | 99.20 | 97.90 | 93.60 | 98.10 | 92.70 | 99.40 | 98.20 | 90.80 |
| | $FS_{rfe}$ | 96.20 | 95.20 | 91.40 | 94.20 | 92.10 | 96.50 | 91.30 | 87.40 |
| | $FS_{bic}$ | 99.20 | 97.80 | 94.00 | 99.00 | 93.80 | **99.60** | **99.60** | 92.90 |

**Table 7.** Performance analysis of classifiers for baseline and its generated feature subsets of adolescents.

|  |  | NB | BG | CART | KNN | C4.5 | KS | SVM | RT |
|---|---|---|---|---|---|---|---|---|---|
| Accuracy | *Baseline* | 93.12 | 80.94 | 82.71 | 88.61 | 89.39 | 87.22 | 95.48 | 80.74 |
|  | $FS_{bor}$ | 94.69 | 81.92 | 83.69 | 92.92 | 89.78 | 93.12 | **95.87** | 78.58 |
|  | $FS_{cfs}$ | 89.91 | 85.08 | 81.04 | 91.93 | 82.66 | 91.53 | 89.51 | 86.69 |
|  | $FS_{rpr}$ | 91.94 | 87.90 | 79.84 | 91.52 | 80.65 | 92.74 | 94.76 | 86.29 |
|  | $FS_{rfe}$ | 87.09 | 75.00 | 78.22 | 88.30 | 81.85 | 88.70 | 89.51 | 78.22 |
|  | $FS_{bic}$ | 92.74 | 87.90 | 81.05 | 91.94 | 81.05 | 91.13 | 93.95 | 89.11 |
| Kappa Stat. | *Baseline* | 89.24 | 61.88 | 65.45 | 77.19 | 78.78 | 74.43 | 90.96 | 61.47 |
|  | $FS_{bor}$ | 89.38 | 63.85 | 67.39 | 85.85 | 79.57 | 86.23 | **91.74** | 57.13 |
|  | $FS_{cfs}$ | 79.78 | 70.14 | 62.08 | 83.84 | 65.31 | 83.00 | 79.02 | 73.31 |
|  | $FS_{rpr}$ | 83.84 | 75.79 | 59.69 | 83.04 | 61.30 | 85.44 | 89.50 | 72.55 |
|  | $FS_{rfe}$ | 74.12 | 49.89 | 56.34 | 76.56 | 63.70 | 77.34 | 78.95 | 56.41 |
|  | $FS_{bic}$ | 85.44 | 75.79 | 62.11 | 83.85 | 62.10 | 82.20 | 87.90 | 78.19 |
| F1-Score | *Baseline* | 93.10 | 80.90 | 82.70 | 88.60 | 89.40 | 87.20 | 95.50 | 80.70 |
|  | $FS_{bor}$ | 94.70 | 81.90 | 83.70 | 92.90 | 89.80 | 93.10 | **95.90** | 78.50 |
|  | $FS_{cfs}$ | 89.20 | 84.60 | 80.70 | 91.50 | 82.30 | 90.70 | 89.30 | 85.70 |
|  | $FS_{rpr}$ | 91.90 | 87.90 | 79.80 | 91.50 | 80.60 | 92.70 | 94.80 | 86.30 |
|  | $FS_{rfe}$ | 86.10 | 73.50 | 76.70 | 87.60 | 81.50 | 87.70 | 88.40 | 77.50 |
|  | $FS_{bic}$ | 92.70 | 87.90 | 81.00 | 91.90 | 81.10 | 91.10 | 94.00 | 89.10 |
| AUROC | *Baseline* | 98.40 | 88.60 | 85.20 | 90.70 | 91.70 | 95.80 | 95.40 | 81.90 |
|  | $FS_{bor}$ | **99.00** | 89.20 | 85.90 | 94.50 | 93.00 | 98.10 | 95.80 | 79.40 |
|  | $FS_{cfs}$ | 93.20 | 96.90 | 82.00 | 95.60 | 81.70 | 97.90 | 89.50 | 87.00 |
|  | $FS_{rpr}$ | 98.60 | 95.70 | 79.60 | 94.60 | 79.80 | 98.30 | 94.70 | 86.30 |
|  | $FS_{rfe}$ | 95.50 | 83.00 | 81.90 | 90.20 | 82.50 | 5.20 | 89.30 | 80.10 |
|  | $FS_{bic}$ | 98.30 | 95.70 | 79.50 | 94.90 | 80.60 | 98.60 | 94.00 | 89.10 |

**Table 8.** Performance analysis of classifiers for baseline and its generated feature subsets of adults.

|  |  | NB | BG | CART | KNN | C4.5 | KS | SVM | RT |
|---|---|---|---|---|---|---|---|---|---|
| Accuracy | *Baseline* | 94.18 | 87.56 | 89.26 | 91.94 | 93.02 | 92.66 | 96.06 | 85.15 |
|  | $FS_{bor}$ | 94.38 | 87.57 | 89.17 | 93.2 | 92.75 | 93.29 | 96.60 | 87.20 |
|  | $FS_{cfs}$ | 95.52 | 87.47 | 89.53 | 96.69 | 93.02 | 97.31 | **99.82** | 87.83 |
|  | $FS_{rpr}$ | 95.34 | 87.50 | 88.99 | 94.99 | 92.66 | 94.45 | 96.42 | 88.90 |
|  | $FS_{rfe}$ | 92.30 | 85.51 | 88.46 | 90.78 | 90.33 | 91.05 | 91.68 | 85.51 |
|  | $FS_{bic}$ | 95.34 | 87.50 | 88.99 | 94.99 | 92.66 | 94.45 | 96.42 | 88.90 |
| Kappa Stat. | *Baseline* | 86.93 | 71.16 | 75.45 | 81.86 | 83.81 | 83.52 | 90.89 | 66.29 |
|  | $FS_{bor}$ | 87.31 | 71.21 | 75.30 | 84.52 | 83.25 | 84.87 | 92.13 | 70.77 |
|  | $FS_{cfs}$ | 89.85 | 71.02 | 76.12 | 92.44 | 83.76 | 93.93 | **99.59** | 72.30 |
|  | $FS_{rpr}$ | 89.46 | 71.21 | 74.90 | 88.60 | 83.05 | 87.52 | 91.67 | 74.49 |
|  | $FS_{rfe}$ | 82.46 | 66.12 | 73.28 | 78.76 | 77.58 | 79.58 | 80.21 | 67.11 |
|  | $FS_{bic}$ | 89.46 | 71.21 | 74.90 | 88.60 | 83.05 | 87.52 | 91.67 | 74.49 |
| F1-Score | $FS_{bor}$ | 94.20 | 87.50 | 89.30 | 92.00 | 93.00 | 92.70 | 96.00 | 85.20 |
|  | $FS_{cfs}$ | 96.70 | 90.80 | 92.80 | 97.60 | 94.90 | 98.00 | **99.90** | 91.00 |
|  | $FS_{rpr}$ | 95.40 | 87.51 | 89.00 | 95.00 | 92.60 | 94.50 | 96.40 | 88.90 |
|  | $FS_{rfe}$ | 94.30 | 89.50 | 91.60 | 93.20 | 93.00 | 93.40 | 94.10 | 89.20 |
|  | $FS_{bic}$ | 95.40 | 87.51 | 89.00 | 95.00 | 92.60 | 94.50 | 96.40 | 88.90 |
| AUROC | *Baseline* | 99.00 | 94.20 | 91.90 | 92.70 | 95.70 | 97.70 | 95.10 | 85.70 |
|  | $FS_{bor}$ | 99.20 | 94.21 | 92.50 | 93.70 | 95.80 | 98.50 | 95.70 | 86.40 |
|  | $FS_{cfs}$ | 99.50 | 94.40 | 92.70 | 99.10 | 96.10 | 99.70 | **99.80** | 87.00 |
|  | $FS_{rpr}$ | 99.50 | 94.30 | 91.60 | 95.70 | 95.80 | 99.00 | 95.20 | 88.20 |
|  | $FS_{rfe}$ | 97.90 | 92.20 | 90.10 | 91.90 | 92.30 | 97.50 | 88.70 | 85.00 |
|  | $FS_{bic}$ | 99.50 | 94.30 | 91.60 | 95.70 | 95.80 | 99.00 | 95.20 | 88.20 |

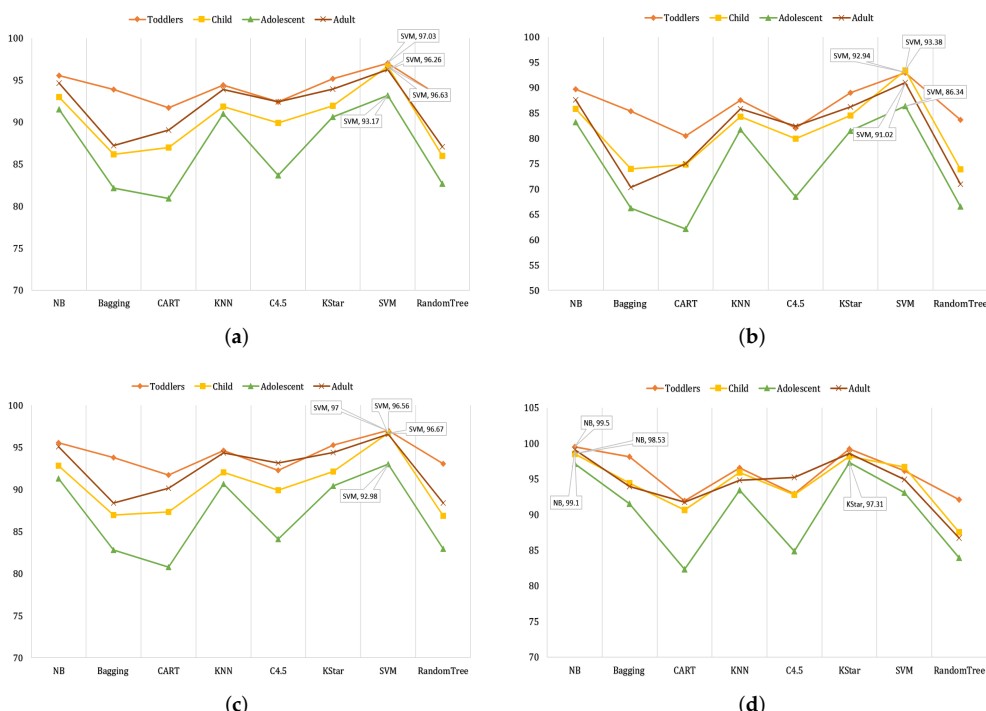

**Figure 3.** The average (**a**) accuracy, (**b**) kappa statistics, (**c**) F1, and (**d**) AUROC of the toddler, child, adolescent, and adult datasets of the classifiers.

*4.3. Result Analysis of Kappa Statistics*

Tables 5–8 show the outcomes of kappa statistics of NB, BG, CART, KNN, C4.5, KS, SVM, and RT for four ASD groups and their feature subsets. In toddlers, the highest kappa (94.87%) was calculated by SVM for $FS_{rpr}$. Regarding the case of children, SVM also provided the highest kappa of 99.21% for $FS_{cfs}$ and $FS_{bic}$. For adolescents, the maximum result (91.74%) was computed by SVM for $FS_{bor}$. In the adult case, the highest kappa (99.59%) was shown by SVM for $FS_{cfs}$. Like the outcomes of accuracy, NB, KS, and KNN showed the closest kappa to detect autism. Besides, the average kappa statistics of the toddler, child, adolescent, and adult datasets are shown in Figure 3b. In this case, SVM again showed the best average kappa in each age group.

*4.4. Result Analysis of F1-Score*

The performance of F-measures in each classifier is shown in Tables 5–8, for ASD baselines and their subsets, respectively. The maximum F1-Score (97.80%) was generated by SVM for $FS_{rpr}$ regarding the toddler dataset. When we observed the findings of the children, SVM also gave the highest f-measure (99.90%) for $FS_{cfs}$ and $FS_{bic}$. In adolescents, the best result (95.90%) was generated by SVM for $FS_{bor}$. For the adult dataset, SVM again provided the highest outcome (96.90%) for $FS_{cfs}$. However, NB, KS, and KNN represented good results, which were almost similar to SVM. The average F-measure of the toddler, child, adolescent, and adult datasets of different classifiers are shown in Figure 3c. Further, SVM provided the average highest results in each age group.

*4.5. Result Analysis of AUROC*

The results of AUROC for the baseline and their feature subsets are shown in Tables 5–8. In toddlers, NB provided the maximum result of 99.70% for $FS_{bor}$. Regarding the children, SVM and KS gave the highest result of 99.60% for $FS_{cfs}$ and $FS_{bic}$, respectively. For adolescents, the maximum result of 99.00% was produced by NB for $FS_{bor}$. In adults, SVM provided a result of 99.80%, which was the best outcome for $FS_{cfs}$. AUROC, NB, SVM, KS, and KNN showed good outcomes, whereas the performance of NB was slightly improved

over others. The average AUROC of the toddler, child, adolescent, and adult datasets of different classifiers are shown in Figure 3d. In this analysis, NB showed the highest average scores to detect autism.

*4.6. Result Analysis of Non-Parametric Statistical Analysis*

In this study, the performance of different classifiers were investigated using non-parametric the Wilcoxon Signed-Rank (WSR) test. We observed pairwise WSR values for toddlers (i.e., Tables 9–12), children (i.e., Tables 13–16), adolescents (i.e., Tables 17–20), and adults (i.e., Tables 21–24). At almost all age levels, SVM is a stable classifier to provide better pairwise statistical significance with other classifiers in order to improve the accuracy, kappa statistics, and f1-measure. Besides, NB sometimes shows better statistical significance for these following metrics. However, this classifier showed better pairwise WSR results with others for AUROC. Some other pairwise outcomes such as C4.5-CART, KNN-CART, KS-BG, KS-CART, KS-KNN, KS-C4.5, KS-RT, CART-RT, KNN-RT, and C4.5-RT showed almost-good outcomes in this test.

**Table 9.** Pairwise WSR test for accuracy of toddlers.

|  | NB | BG | CART | KNN | C4.5 | KS | SVM | RT |
|---|---|---|---|---|---|---|---|---|
| NB |  | 0.0211 ** | 0.0211 ** | 0.0592 * | 0.0179 ** | 0.2643 | 1.0000 | 0.0211 ** |
| BG | 1.0000 |  | 0.0211 ** | 1.0000 | 0.0211 ** | 1.0000 | 1.0000 | 0.0592 * |
| CART | 1.0000 | 1.0000 |  | 1.0000 | 1.0000 | 1.0000 | 1.0000 | 1.0000 |
| KNN | 1.0000 | 0.1422 | 0.0179 ** |  | 0.0211 ** | 1.0000 | 1.0000 | 0.1787 |
| C4.5 | 1.0000 | 1.0000 | 0.0211 ** | 1.0000 |  | 1.0000 | 1.0000 | 1.0000 |
| KS | 1.0000 | 0.0211 ** | 0.0211 ** | 0.0256 ** | 0.0211 ** |  | 1.0000 | 0.1787 |
| SVM | 0.0211 ** | 0.0211 ** | 0.1787 | 0.1787 | 0.0211 ** | 0.0211 ** |  | 0.0211 ** |
| RT | 1.0000 | 1.0000 | 0.0211 ** | 1.0000 | 0.0408 ** | 1.0000 | 1.0000 |  |

Note: Significant at 5% = **, 10% = *.

**Table 10.** Pairwise WSR test for kappa statistics of toddlers.

|  | NB | BG | CART | KNN | C4.5 | KS | SVM | RT |
|---|---|---|---|---|---|---|---|---|
| NB |  | 0.0211 ** | 0.0211 ** | 0.0592 * | 0.0211 ** | 0.4017 | 1.0000 | 0.0211 ** |
| BG | 1.0000 |  | 0.0211 ** | 1.0000 | 0.0211 ** | 1.0000 | 1.0000 | 0.0592 * |
| CART | 1.0000 | 1.0000 |  | 1.0000 | 1.0000 | 1.0000 | 1.0000 | 1.0000 |
| KNN | 1.0000 | 0.0592 * | 0.0211 ** |  | 0.0211 ** | 1.0000 | 1.0000 | 0.0211 ** |
| C4.5 | 1.0000 | 1.0000 | 0.0211 | 1.0000 |  | 1.0000 | 1.0000 | 1.0000 |
| KS | 1.0000 | 0.0211 ** | 0.0211 ** | 0.0211 ** | 0.0211 ** |  | 1.0000 | 0.0211 ** |
| SVM | 0.0211 ** | 0.0211 ** | 0.0211 ** | 0.0211 ** | 0.0211 ** | 0.0211 ** |  | 0.0211 ** |
| RT | 1.0000 | 1.0000 | 0.0211 | 1.0000 | 0.0360 ** | 1.0000 | 1.0000 |  |

Note: Significant at 5% = **, 10% = *.

**Table 11.** Pairwise WSR test for F1-measure of toddlers.

|  | NB | BG | CART | KNN | C4.5 | KS | SVM | RT |
|---|---|---|---|---|---|---|---|---|
| NB |  | 0.0179 ** | 0.0179 ** | 0.0592 * | 0.0100 *** | 0.4511 | 1.0000 | 0.0179 ** |
| BG | 1.0000 |  | 0.0179 ** | 1.0000 | 0.0179 ** | 1.0000 | 1.0000 | 0.5917 |
| CART | 1.0000 | 1.0000 |  | 1.0000 | 1.0000 | 1.0000 | 1.0000 | 1.0000 |
| KNN | 1.0000 | 0.0591 * | 0.0211 ** |  | 0.0211 ** | 1.0000 | 1.0000 | 0.0211 ** |
| C4.5 | 1.0000 | 1.0000 | 0.0179 ** | 1.0000 |  | 1.0000 | 1.0000 | 1.0000 |
| KS | 1.0000 | 0.0211 ** | 0.0211 ** | 0.0256 ** | 0.0211 ** |  | 1.0000 | 0.0211 ** |
| SVM | 0.0211 ** | 0.0211 ** | 0.0211 ** | 0.0211 ** | 0.1787 | 0.0211 ** |  | 0.0211 ** |
| RT | 1.0000 | 1.0000 | 0.0211 ** | 1.0000 | 0.0408 ** | 1.0000 | 1.0000 |  |

Note: Significant at 1% = ***, 5% = **, 10% = *.

**Table 12.** Pairwise WSR test for AUROC of toddlers.

|  | NB | BG | CART | KNN | C4.5 | KS | SVM | RT |
|---|---|---|---|---|---|---|---|---|
| NB |  | 0.0149 ** | 0.0179 ** | 0.0211 ** | 0.0211 ** | 0.0592 * | 0.0211 ** | 0.0211 ** |
| BG | 1.0000 |  | 0.0179 ** | 0.0360 ** | 0.0179 ** | 1.0000 | 0.0179 ** | 0.0211 ** |
| CART | 1.0000 | 1.0000 |  | 1.0000 | 1.0000 | 1.0000 | 1.0000 | 0.3326 |
| KNN | 1.0000 | 1.0000 | 0.0211 ** |  | 0.0211 ** | 1.0000 | 0.9143 | 0.0179 ** |
| C4.5 | 1.0000 | 1.0000 | 0.0179 ** | 1.0000 |  | 1.0000 | 1.0000 | 0.0672 * |
| KS | 1.0000 | 0.0211 ** | 0.0179 ** | 0.0211 ** | 0.0179 ** |  | 0.0211 ** | 0.0211 ** |
| SVM | 1.0000 | 1.0000 | 0.0211 ** | 0.9143 | 0.0179 ** | 1.0000 |  | 0.0313 ** |
| RT | 1.0000 | 1.0000 | 1.0000 | 1.0000 | 1.0000 | 1.0000 | 1.0000 |  |

Note: Significant at 5% = **, 10% = *.

**Table 13.** Pairwise WSR test for accuracy of child.

|  | NB | BG | CART | KNN | C4.5 | KS | SVM | RT |
|---|---|---|---|---|---|---|---|---|
| NB |  | 0.0100 *** | 0.0179 ** | 0.5183 | 0.0310 ** | 0.8295 | 1.0000 | 0.0211 ** |
| BG | 1.0000 |  | 1.0000 | 1.0000 | 1.0000 | 1.0000 | 1.0000 | 0.6750 |
| CART | 1.0000 | 0.2817 |  | 1.0000 | 1.0000 | 1.0000 | 1.0000 | 0.5294 |
| KNN | 1.0000 | 0.0179 ** | 0.0179 ** |  | 0.0592 * | 1.0000 | 1.0000 | 0.0211 ** |
| C4.5 | 1.0000 | 0.2945 | 0.0310 ** | 1.0000 |  | 1.0000 | 1.0000 | 0.2945 |
| KS | 1.0000 | 0.0179 ** | 0.0179 ** | 0.4024 | 0.0592 ** |  | 1.0000 | 0.0211 ** |
| SVM | 0.0149 ** | 0.0179 ** | 0.0179 ** | 0.0179 ** | 0.0179 ** | 0.0179 ** |  | 0.0211 ** |
| RT | 1.0000 | 1.0000 | 1.0000 | 1.0000 | 1.0000 | 1.0000 | 1.0000 |  |

Note: Significant at 1% = ***, 5% = **, 10% = *.

**Table 14.** Pairwise WSR Test for kappa statistics of children.

|  | NB | BG | CART | KNN | C4.5 | KS | SVM | RT |
|---|---|---|---|---|---|---|---|---|
| NB |  | 0.0100 *** | 0.0179 ** | 0.5183 | 0.0256 ** | 0.8295 | 1.0000 | 0.0179 ** |
| BG | 1.0000 |  | 1.0000 | 1.0000 | 1.0000 | 1.0000 | 1.0000 | 0.6668 |
| CART | 1.0000 | 0.2817 |  | 1.0000 | 1.0000 | 1.0000 | 1.0000 | 0.8295 |
| KNN | 1.0000 | 0.0179 ** | 0.0179 ** |  | 0.0527 * | 1.0000 | 1.0000 | 0.0179 ** |
| C4.5 | 1.0000 | 0.2817 | 0.0256 ** | 1.0000 |  | 1.0000 | 1.0000 | 0.2817 |
| KS | 1.0000 | 0.0179 ** | 0.0179 ** | 0.3891 | 0.0527 * |  | 1.0000 | 0.0179 ** |
| SVM | 0.0149 ** | 0.0179 ** | 0.0179 ** | 0.0179 ** | 0.0179 ** | 0.0179 ** |  | 0.0179 ** |
| RT | 1.0000 | 1.0000 | 1.0000 | 1.0000 | 1.0000 | 1.0000 | 1.0000 |  |

Note: Significant at 1% = ***, 5% = **, 10% = *.

**Table 15.** Pairwise WSR test for F1-Measure of children.

|  | NB | BG | CART | KNN | C4.5 | KS | SVM | RT |
|---|---|---|---|---|---|---|---|---|
| NB |  | 0.0179 ** | 0.0179 ** | 0.5294 | 0.0310 ** | 0.6668 | 1.0000 | 0.0211 ** |
| BG | 1.0000 |  | 0.0850 * | 1.0000 | 1.0000 | 1.0000 | 1.0000 | 0.6668 |
| CART | 1.0000 | 1.0000 |  | 1.0000 | 1.0000 | 1.0000 | 1.0000 | 1.0000 |
| KNN | 1.0000 | 0.0211 ** | 0.0211 ** |  | 0.0592 * | 1.0000 | 1.0000 | 0.0211 ** |
| C4.5 | 1.0000 | 0.2817 | 0.0211 ** | 1.0000 |  | 1.0000 | 1.0000 | 0.2945 |
| KS | 1.0000 | 0.0211 ** | 0.0179 ** | 0.5904 | 0.0935 * |  | 1.0000 | 0.0179 ** |
| SVM | 0.0211 ** | 0.0211 ** | 0.0179 ** | 0.0211 ** | 0.0211 ** | 0.0211 ** |  | 0.0179 ** |
| RT | 1.0000 | 1.0000 | 0.1318 | 1.0000 | 1.0000 | 1.0000 | 1.0000 |  |

Note: Significant at 5% = **, 10% = *.

**Table 16.** Pairwise WSR Test for AUROC of Child.

|  | **NB** | **BG** | **CART** | **KNN** | **C4.5** | **KS** | **SVM** | **RT** |
|---|---|---|---|---|---|---|---|---|
| NB |  | 0.0179 ** | 0.0100 *** | 0.0179 ** | 0.0211 ** | 0.8295 | 0.0850 * | 0.0211 ** |
| BG | 1.0000 |  | 0.0179 ** | 1.0000 | 0.1617 | 1.0000 | 1.0000 | 0.0211 ** |
| CART | 1.0000 | 1.0000 |  | 1.0000 | 1.0000 | 1.0000 | 1.0000 | 1.0000 |
| KNN | 1.0000 | 0.0527 * | 0.0179 ** |  | 0.0360 ** | 1.0000 | 1.0000 | 0.0211 ** |
| C4.5 | 1.0000 | 1.0000 | 0.0211 ** | 1.0000 |  | 1.0000 | 1.0000 | 0.0211 ** |
| KS | 1.0000 | 0.0179 ** | 0.0179 ** | 0.0179 ** | 0.0211 ** |  | 0.0464 ** | 0.0211 ** |
| SVM | 1.0000 | 0.1318 | 0.0179 ** | 0.1964 | 0.0360 ** | 1.0000 |  | 0.0211 ** |
| RT | 1.0000 | 1.0000 | 0.0935 * | 1.0000 | 1.0000 | 1.0000 | 1.0000 |  |

Note: Significant at 1% = ***, 5% = **, 10% = *.

**Table 17.** Pairwise WSR test for accuracy of adolescents.

|  | **NB** | **BG** | **CART** | **KNN** | **C4.5** | **KS** | **SVM** | **RT** |
|---|---|---|---|---|---|---|---|---|
| NB |  | 0.0179 ** | 0.0100 *** | 0.0179 ** | 0.0211 ** | 0.8295 | 0.0850 * | 0.0211 ** |
| BG | 1.0000 |  | 0.0179 ** | 1.0000 | 0.1617 | 1.0000 | 1.0000 | 0.0211 ** |
| CART | 1.0000 | 1.0000 |  | 1.0000 | 1.0000 | 1.0000 | 1.0000 | 1.0000 |
| KNN | 1.0000 | 0.0527 * | 0.0179 ** |  | 0.0360 ** | 1.0000 | 1.0000 | 0.0211 ** |
| C4.5 | 1.0000 | 1.0000 | 0.0211 ** | 1.0000 |  | 1.0000 | 1.0000 | 0.0211 ** |
| KS | 1.0000 | 0.0179 ** | 0.0179 ** | 0.0179 ** | 0.0211 ** |  | 0.0464 * | 0.0211 ** |
| SVM | 1.0000 | 0.1318 | 0.0179 ** | 0.1964 | 0.0360 ** | 1.0000 |  | 0.0211 ** |
| RT | 1.0000 | 1.0000 | 0.0935 * | 1.0000 | 1.0000 | 1.0000 | 1.0000 |  |

Note: Significant at 1% = ***, 5% = **, 10% = *.

**Table 18.** Pairwise WSR test for kappa statistics of adolescents.

|  | **NB** | **BG** | **CART** | **KNN** | **C4.5** | **KS** | **SVM** | **RT** |
|---|---|---|---|---|---|---|---|---|
| NB |  | 0.0211 ** | 0.0211 ** | 0.5294 | 0.0211 ** | 0.5183 | 1.0000 | 0.0211 ** |
| BG | 1.0000 |  | 0.2945 | 1.0000 | 1.0000 | 1.0000 | 1.0000 | 0.8339 |
| CART | 1.0000 | 1.0000 |  | 1.0000 | 1.0000 | 1.0000 | 1.0000 | 1.0000 |
| KNN | 1.0000 | 0.0211 ** | 0.0211 ** |  | 0.0360 ** | 0.5294 | 1.0000 | 0.0211 ** |
| C4.5 | 1.0000 | 0.4017 | 0.0360 ** | 1.0000 |  | 1.0000 | 1.0000 | 0.6750 |
| KS | 1.0000 | 0.0211 ** | 0.0211 ** | 1.0000 | 0.0360 ** |  | 1.0000 | 0.0211 ** |
| SVM | 0.0360 | 0.0211 ** | 0.0211 ** | 0.0935 | 0.0211 ** | 0.0592 * |  | 0.0211 ** |
| RT | 1.0000 | 1.0000 | 0.2084 | 1.0000 | 1.0000 | 1.0000 | 1.0000 |  |

Note: Significant at 5% = **, 10% = *.

**Table 19.** Pairwise WSR test for F1-Measure of adolescents.

|  | **NB** | **BG** | **CART** | **KNN** | **C4.5** | **KS** | **SVM** | **RT** |
|---|---|---|---|---|---|---|---|---|
| NB |  | 0.0211 ** | 0.0211 ** | 0.5294 | 0.0211 ** | 0.3488 | 1.0000 | 0.0211 ** |
| BG | 1.0000 |  | 0.2817 | 1.0000 | 1.0000 | 1.0000 | 1.0000 | 1.0000 |
| CART | 1.0000 | 1.0000 |  | 1.0000 | 1.0000 | 1.0000 | 1.0000 | 1.0000 |
| KNN | 1.0000 | 0.0211 ** | 0.0211 ** |  | 0.0360 ** | 0.5294 | 1.0000 | 0.0211 ** |
| C4.5 | 1.0000 | 0.2945 | 0.0211 ** | 1.0000 |  | 1.0000 | 1.0000 | 0.5294 |
| KS | 1.0000 | 0.0211 ** | 0.0211 ** | 1.0000 | 0.0360 ** |  | 1.0000 | 0.0211 ** |
| SVM | 0.0211 ** | 0.0211 ** | 0.0211 ** | 0.0935 * | 0.0211 ** | 0.0592 * |  | 0.0211 ** |
| RT | 1.0000 | 0.8339 | 0.2945 | 1.0000 | 1.0000 | 1.0000 | 1.0000 |  |

Note: Significant at 5% = **, 10% = *.

**Table 20.** Pairwise WSR test for AUROC of adolescents.

|        | NB       | BG        | CART      | KNN       | C4.5      | KS        | SVM       | RT        |
|--------|----------|-----------|-----------|-----------|-----------|-----------|-----------|-----------|
| NB     |          | 0.0935 *  | 0.0211 ** | 0.0360 ** | 0.0211 ** | 0.0360 ** | 0.0211 ** | 0.0211 ** |
| BG     | 1.0000   |           | 0.0211 ** | 1.0000    | 0.0208 ** | 1.0000    | 1.0000    | 0.0211 ** |
| CART   | 1.0000   | 1.0000    |           | 1.0000    | 1.0000    | 1.0000    | 1.0000    | 1.0000    |
| KNN    | 1.0000   | 0.2945    | 0.0211 ** |           | 0.0360 ** | 1.0000    | 0.8295    | 0.0211 ** |
| C4.5   | 1.0000   | 1.0000    | 0.0592 *  | 1.0000    |           | 1.0000    | 1.0000    | 0.6750    |
| KS     | 1.0000   | 0.2945    | 0.2945    | 0.2945    | 0.2945    |           | 0.2945    | 0.2945    |
| SVM    | 1.0000   | 0.6750    | 0.0211 ** | 1.0000    | 0.0211 ** | 1.0000    |           | 0.0211 ** |
| RT     | 1.0000   | 1.0000    | 0.4017    | 1.0000    | 1.0000    | 1.0000    | 1.0000    |           |

Note: Significant at 5% = **, 10% = *.

**Table 21.** Pairwise WSR test for accuracy of adults.

|        | NB        | BG        | CART      | KNN       | C4.5      | KS        | SVM    | RT        |
|--------|-----------|-----------|-----------|-----------|-----------|-----------|--------|-----------|
| NB     |           | 0.0179 ** | 0.0179 ** | 0.0850 *  | 0.0179 ** | 0.2817    | 1.0000 | 0.0179 ** |
| BG     | 1.0000    |           | 1.0000    | 1.0000    | 1.0000    | 1.0000    | 1.0000 | 1.0000    |
| CART   | 1.0000    | 0.0179 ** |           | 1.0000    | 1.0000    | 1.0000    | 1.0000 | 0.0179 ** |
| KNN    | 1.0000    | 0.0179 ** | 0.0179 ** |           | 0.0767 *  | 1.0000    | 1.0000 | 0.0179 ** |
| C4.5   | 1.0000    | 0.0179 ** | 0.0179 ** | 1.0000    |           | 1.0000    | 1.0000 | 0.0179 ** |
| KS     | 1.0000    | 0.0179 ** | 0.0179 ** | 0.3891    | 0.0313 ** |           | 1.0000 | 0.0179 ** |
| SVM    | 0.0313 ** | 0.0179 ** | 0.0179 ** | 0.0179 ** | 0.0179 ** | 0.0179 ** |        | 0.0179 ** |
| RT     | 1.0000    | 0.7802    |           | 1.0000    | 1.0000    | 1.0000    | 1.0000 |           |

Note: Significant at 5% = **, 10% = *.

**Table 22.** Pairwise WSR test for kappa statistics of adults.

|        | NB        | BG        | CART      | KNN       | C4.5      | KS        | SVM    | RT        |
|--------|-----------|-----------|-----------|-----------|-----------|-----------|--------|-----------|
| NB     |           | 0.0179 ** | 0.0179 ** | 0.0850 *  | 0.0179 ** | 0.2817    | 1.0000 | 0.0179 ** |
| BG     | 1.0000    |           | 1.0000    | 1.0000    | 1.0000    | 1.0000    | 1.0000 | 1.0000    |
| CART   | 1.0000    | 0.0179 ** |           | 1.0000    | 1.0000    | 1.0000    | 1.0000 | 0.0179 ** |
| KNN    | 1.0000    | 0.0179 ** | 0.0179 ** |           | 0.0767 *  | 1.0000    | 1.0000 | 0.0179 ** |
| C4.5   | 1.0000    | 0.0179 ** | 0.0179 ** | 1.0000    |           | 1.0000    | 1.0000 | 0.0179 ** |
| KS     | 1.0000    | 0.0179 ** | 0.0179 ** | 0.3891    | 0.0313 ** |           | 1.0000 | 0.0179 ** |
| SVM    | 0.0313    | 0.0179 ** | 0.0179 ** | 0.0179 ** | 0.0179 ** | 0.0179 ** |        | 0.0179 ** |
| RT     | 1.0000    | 0.3891    | 1.0000    | 1.0000    | 1.0000    | 1.0000    | 1.0000 |           |

Note: Significant at 5% = **, 10% = *.

**Table 23.** Pairwise WSR test for F1-Measure of adults.

|        | NB        | BG        | CART      | KNN       | C4.5      | KS        | SVM    | RT        |
|--------|-----------|-----------|-----------|-----------|-----------|-----------|--------|-----------|
| NB     |           | 0.0256 ** | 0.0256 ** | 0.1629    | 0.0256 ** | 0.2643    | 1.0000 | 0.0256 ** |
| BG     | 1.0000    |           | 1.0000    | 1.0000    | 1.0000    | 1.0000    | 1.0000 | 1.0000    |
| CART   | 1.0000    | 0.0256 ** |           | 1.0000    | 1.0000    | 1.0000    | 1.0000 | 0.0256 ** |
| KNN    | 1.0000    | 0.0256 ** | 0.0256 ** |           | 0.0940    | 1.0000    | 1.0000 | 0.0256 ** |
| C4.5   | 1.0000    | 0.0256 ** | 0.0256 ** | 1.0000    |           | 1.0000    | 1.0000 | 0.0256 ** |
| KS     | 1.0000    | 0.0256 ** | 0.0256 ** | 0.7802    | 0.0507 *  |           | 1.0000 | 0.0256 ** |
| SVM    | 0.0507 *  | 0.0256 ** | 0.0256 ** | 0.0256 ** | 0.0256 ** | 0.0126 ** |        | 0.0256 ** |
| RT     | 1.0000    | 0.7802    | 1.0000    | 1.0000    | 1.0000    | 1.0000    | 1.0000 |           |

Note: Significant at 5% = **, 10% = *.

**Table 24.** Pairwise WSR test for AUROC of adults.

|  | NB | BG | CART | KNN | C4.5 | KS | SVM | RT |
|---|---|---|---|---|---|---|---|---|
| NB |  | 0.0179 ** | 0.0179 ** | 0.0179 ** | 0.0149 ** | 0.0313 ** | 0.0313 ** | 0.0179 ** |
| BG | 1.0000 |  | 0.0179 ** | 1.0000 | 1.0000 | 1.0000 | 1.0000 | 0.0179 ** |
| CART | 1.0000 | 1.0000 |  | 1.0000 | 1.0000 | 1.0000 | 1.0000 | 0.0179 ** |
| KNN | 1.0000 | 0.5183 | 0.0179 ** |  | 1.0000 | 1.0000 | 1.0000 | 0.0179 ** |
| C4.5 | 1.0000 | 0.0100 *** | 0.0179 ** | 0.2236 |  | 1.0000 | 0.2817 | 0.0179 ** |
| KS | 1.0000 | 0.0149 ** | 0.0100 *** | 0.0179 ** | 0.0179 ** |  | 0.0313 ** | 0.0179 ** |
| SVM | 1.0000 | 0.1964 | 0.0313 ** | 0.6668 | 1.0000 | 1.0000 |  | 0.0179 ** |
| RT | 1.0000 | 1.0000 | 1.0000 | 1.0000 | 1.0000 | 1.0000 | 1.0000 |  |

Note: Significant at 1% = ***, 5% = **.

*4.7. Exploring Significant Feature Sets and Discriminatory Factors of Individual Age Groups*

For toddler datasets, SVM represented their maximum result in all metrics except AUROC for $FS_{rpr}$. When we considered the value of AUROC, NB showed the best result for $FS_{rpr}$. Then, SVM also showed their highest outcomes for all metrics for $FS_{cfs}$ and $FS_{bic}$ in child datasets. In addition, KS provided the best AUROC for $FS_{cfs}$ and $FS_{bic}$. In adolescent datasets, SVM showed the best accuracy, kappa statistics, and f1-score for $FS_{bor}$. However, NB represented the top AUROC for $FS_{cfs}$. In adult datasets, SVM also produced the highest findings in all metrics for $FS_{cfs}$. Therefore, it was found that the RIPPER algorithm for toddler, the Boruta algorithm for adolescent, and the CFS method for child and adult datasets were responsible to generate high results in the classification process.

Figure 2 depicts the ranks of SHAP values of toddler's RIPPER subset, adolescent's Boruta subset, as well as the subsets of CFS child and adult datasets. The assessment of these values was performed with the best-performing SVM for these individual subsets. For the RIPPER subset of toddlers, the most important discriminatory features were "age group identification," "Character's Intention (A7)," "Following relook (A6)," "Pretending Capability (A5)," and "Pointing to the interest (A4)" to detect autism at an early stage. For the CFS subset of children, "Back to the activities (A4)," "Making Friends (A10)," "Finding the character's intention (A7)," "Understanding someone's feeling (A9)," and "Noticing sound (A1)" were the most crucial discriminating factors. Besides, "Social activity (A6)," "Indicating toy (A3)," "Pretending Capability (A5)," "Return to work (A4)," and "Developing Relationships (A10)" were the most significant discriminating factors for the Boruta subset of adolescents. In the CFS subset of adults, "Understanding someone's feelings (A9)," "Pretending Capability (A5)," "Social activity (A6)," "Track activities (A3)," and "Return to work (A4)" are the major discriminating factors to detect autism.

**5. Discussion and Conclusions**

In this study, we proposed a machine learning model that analyzed ASD datasets of individual age groups (toddlers, children, adolescents, and adults) to detect autism at an early stage. This model used different feature selection methods like Boruta, CFS, RIPPER, and RFE to generate feature subsets. Then, NB, BG, CART, KNN, C4.5, KS, SVM, and RT were employed to classify autism at an early stage. When we evaluated the performance of them, SVM was the most stable classifier to explore the best result for different age groups, respectively. Along with SVM, KS, KNN, and NB showed better results to identify autism, respectively. On the other hand, individual classifiers presented their best performance in $FS_{rpr}$ for toddlers, $FS_{bor}$ for adolescents, $FS_{cfs}$ for children and adults, and $FS_{bic}$ for children.

Table 25 shows the comparison of the proposed model with related previous studies. Most of the existing works were worked with version-1 ASD datasets of Thabtah et al. [6]. Besides, they did not properly focus on early detection of autism for toddlers and adolescents. Some works occurred with version-2 adolescent and adult datasets, whereas most of them did not conduct those works with more samples and machine learning approaches. Thabtah et al. [14] used CHI and IG feature ranking methods into primary adolescent and

adult datasets to produce the most significant feature subsets. Then, they implemented logistic regression into primary datasets and subsets and observed the highest 99.91% accuracy for the adolescent dataset and a 97.58% accuracy for the adult dataset. In this work, we investigated all version-2 datasets of different age groups of Thabtah et al. [14] where SVM achieved a 97.82% accuracy with $FS_{rpr}$ for the toddler dataset, a 99.61% accuracy with $FS_{cfs}$ and $FS_{bic}$ for the child dataset, a 95.87% accuracy with $FS_{bor}$ for the adolescent dataset, and a 96.82% accuracy for the adult dataset for $FS_{cfs}$. Besides, we applied numerous feature selection and classification methods to justify these baselines and its feature sets more efficiently. In addition, a post hoc statistical significant test and SHAP interpretation method was used to evaluate deeply the outcomes of the proposed models. These types of evaluations did not properly occur in most of the existing works.

**Table 25.** Comparison of proposed model with other previous studies.

| Dataset | Version | Feature Reduction | Reference | Accuracy (%) | Kappa. (%) | F1 (%) | AUROC (%) |
|---|---|---|---|---|---|---|---|
| Toddler | v1 | No | [57] | - | - | - | - |
| | v1 | No | [58] | - | - | - | - |
| | v2 | No | [14] | - | - | - | - |
| | v2 | Yes | Proposed Model | 97.82 | 94.87 | 97.80 | 99.70 |
| Child | v1 | No | [57] | - | - | - | - |
| | v1 | No | [58] | 98.62 | 98.60 | - | - |
| | v2 | No | [14] | - | - | - | - |
| | v2 | Yes | Proposed Model | 99.61 | 99.21 | 99.60 | 99.60 |
| Adolescent | v1 | No | [57] | - | - | - | - |
| | v1 | No | [58] | - | - | - | - |
| | v2 | No | [14] | 99.91 | - | - | - |
| | v2 | Yes | Proposed Model | 95.87 | 91.74 | 95.90 | 99.00 |
| Adult | v1 | No | [57] | 91.74 | - | - | - |
| | v1 | No | [58] | 99.73 | 99.38 | - | - |
| | v2 | No | [14] | 97.58 | - | - | - |
| | v2 | Yes | Proposed Model | 99.82 | 99.59 | 99.90 | 99.80 |

In conclusions, the proposed machine learning framework was used to produce more-accurate and efficient results for early detection of ASD at individual age groups. Since diagnosing ASD traits is an expensive and time-consuming procedure, it is often postponed due to the difficulty of detecting autism in children and adolescents. In this process, machine learning models are efficient to detect autism at an early stage very efficiently. However, this model was not trained with various multivariate/dimensional datasets and explored significant attributes. In the future, we will integrate this framework with advanced technologies and develop a more-efficient ASD diagnosis system. This system will be applicable for the preliminary diagnosis of ASD at an early stage with low costs.

**Author Contributions:** Conceptualization, M.B. and M.H.A.; methodology and software, M.B.; validation and formal analysis, M.H.A., M.S.S. and K.F.H.; resources, M.B. and M.S.S.; data curation, M.B.; writing—original draft preparation, M.B. and M.S.S., writing—review and editing, M.B., M.S.S., K.F.H. and M.A.M.; visualization, M.B.; supervision, M.A.M. All authors have read and agreed to the published version of the manuscript.

**Funding:** This research received no external funding.

**Informed Consent Statement:** Not applicable.

**Data Availability Statement:** The data used in this paper is available in the references in Section 3.1.

**Conflicts of Interest:** The authors declare no conflict of interest.

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
