# Peer review of "Efficient Machine Learning Models for Early Stage Detection of Autism Spectrum Disorder"

_algorithms, doi:10.3390/a15050166_

Round 1
Reviewer 1 Report
The authors used a variety of ASD datasets and, as stated in the paper, only used individual classifiers to classify the datasets. However, there is no clear methodology because Figure 1 is so broad. All of the used classifiers have been used before, and no methodology has been used to determine if there is anything new in the methodology that can be deemed a contribution.
Author Response
The authors used a variety of ASD datasets and, as stated in the paper, only used individual classifiers to classify the datasets. However, there is no clear methodology because Figure 1 is so broad. All of the used classifiers have been used before, and no methodology has been used to determine if there is anything new in the methodology that can be deemed a contribution.
Response: We admire the reviewer for raising some aspects in this study. There were happened many works such kinds of things. In revised manuscript, we modified Figure 1 and this model represents some contributions to detect autism. In this study, we investigated autistic instances of different age levels where individual characteristics were identified using existing classifiers and other methods.
- There were used many feature selection methods to recognize which feature subsets are best in the individual age levels.
- We implemented post hoc statistical test to justify the performance of classifiers
- We interpreted the classification results of using explainable AI technique and extract various characteristics of autistic people.

Reviewer 2 Report
Medical decision support is an important task of computer science and artificial intelligence. The paper submitted for review fits well into such a demand. It proposes machine learning methods for detecting autistic disorders.
The paper is written well and clearly - both in the layer of introduction to the topic and in the layer of descriptions of experiments. The authors have clearly presented the motives for undertaking this work.
The aspects related to the selection of classifiers and the metrics determining the quality of classification are also correctly presented.
Technical comments:
-no affiliation of the last author of the publication
- Table 2 can be shortened (simplified) because the last rows (A1-A10) of this table can be aggregated.
Substantive comments:
- Table 11 shows a comparison of the proposed method with others from the literature. This comparison makes sense if the studies used exactly the same databases and examined exactly the same number of cases. It does not appear from the article that this was the case. Please comment on this issue.
In Tables 6-9, there are numerical records (in columns) that, from a statistical point of view, do not necessarily differ. There should be a statistical analysis (e.g., with the Wicoxon test) for these data that the results are statistically different from each other.
In the summary of the paper, the authors should clearly state the recommendations for using the method with a clear distinction between toddlers, children, adolescents, and adults. Currently, the summary only states which technique and which subset of features produces acceptable results.
Author Response
Medical decision support is an important task of computer science and artificial intelligence. The paper submitted for review fits well into such a demand. It proposes machine learning methods for detecting autistic disorders. The paper is written well and clearly - both in the layer of introduction to the topic and in the layer of descriptions of experiments. The authors have clearly presented the motives for undertaking this work. The aspects related to the selection of classifiers and the metrics determining the quality of classification are also correctly presented
Response: We thank the reviewer for the positive comments. Your valuable comments have increased the quality of this work.
Technical comments:
-no affiliation of the last author of the publication
Response: This mistake has been fixed in the revised manuscript.
- Table 2 can be shortened (simplified) because the last rows (A1-A10) of this table can be aggregated.
Response: Thanks for your suggestion. According to your comments, Table 2 is simplified in the revised manuscript.
Substantive comments:
- Table 11 shows a comparison of the proposed method with others from the literature. This comparison makes sense if the studies used exactly the same databases and examined exactly the same number of cases. It does not appear from the article that this was the case. Please comment on this issue.
Response: We agree with reviewer regarding this point. Table 11 compared the proposed model to various recent works and demonstrates which model is better. The majority of previous studies were used version-1 ASD datasets of Thabtah et al [14] where they did not appropriately focus on toddler and adolescents. Some of the studies were only investigated with version-2 adolescent and adult of same datasets. Therefore, we investigated ASD in the Toddler, Child, Adolescent, and Adult using version 1 and version 2 datasets of Thabtah et al. Then, we achieved (97.82%) accuracy by SVM for toddler dataset, (99.61%) for the child dataset, (95.87%) for the adolescent dataset and (96.82%) accuracy for the adult dataset. When we compared with other works, we observed that proposed model showed the highest results to predict ASD.
In Tables 6-9, there are numerical records (in columns) that, from a statistical point of view, do not necessarily differ. There should be a statistical analysis (e.g., with the Wilcoxon test) for these data that the results are statistically different from each other.
Response: Thank you for your concern. We give some post hoc statistical analysis in the revised manuscript.
In the summary of the paper, the authors should clearly state the recommendations for using the method with a clear distinction between toddlers, children, adolescents, and adults. Currently, the summary only states which technique and which subset of features produces acceptable results.
Response: We agree with the reviewer regarding this point. In the update manuscript, we clearly recommended suitable technique and feature subsets that provided best outcomes for toddlers, children, adolescents, and adults clearly.

Round 2
Reviewer 2 Report
Thanks for improving the manuscript. All my przeviously statet comments have beenacceped and now paper sounds better.Paper can be published.